# Rapid spread of the SARS-CoV-2 Omicron XDR lineage derived from recombination between XBB and BA.2.86 subvariants circulating in Brazil in late 2023

Ighor Arantes,[1] Kimihito Ito,[2] Marcelo Gomes,[3] Felipe Cotrim de Carvalho,[4] Walquiria Aparecida Ferreira de Almeida,[4] Ricardo Khouri,[5] Fabio Miyajima,[6] Gabriel Luz Wallau,[7,8] Felipe Gomes Naveca,[1,9] Elisa Cavalcante Pereira,[10] COVID-19 Fiocruz Genomic Surveillance Network, Marilda Mendonça Siqueira,[10] Paola Cristina Resende,[10] Gonzalo Bello[1]

**ABSTRACT**  Recombination plays a crucial role in the evolution of SARS-CoV-2. The Omicron XBB* recombinant lineages are a noteworthy example, as they have been the dominant SARS-CoV-2 variant worldwide in the first half of 2023. Since November 2023, a new recombinant lineage between Omicron subvariants XBB and BA.2.86, designated XDR, has been detected mainly in Brazil. In this study, we reconstructed the spatiotemporal dynamics and estimated the absolute and relative transmissibility of the XDR lineage. The XDR lineage displayed a recombination breakpoint in the ORF1a-coding region, and the most closely related sequences to the 5′ and 3′ ends of the recombinant correspond to JD.1.1 and JN.1.1 lineages, respectively. The first XDR sequences were detected in November 2023 in the Northeastern Brazilian region, and their prevalence rapidly surged from <1% to 25% by February 2024. The Bayesian phylogeographic analysis supports that the XDR lineage likely emerged in the Northeastern Brazilian region around late October 2023 and rapidly disseminated within and outside Brazilian borders from mid-November onward. The median effective reproductive number of the XDR lineage in Brazil during the initial expansion phase was estimated to be around 1.5, and the average relative instantaneous reproduction numbers of XDR and JN* lineages were estimated to be 1.37 and 1.29 higher than that of co-circulating XBB* lineages. In summary, these findings support that the recombinant lineage XDR arose in the Northeastern Brazilian region in October 2023, shortly after the first detection of JN.1 sequences in the country. In Brazil, the XDR lineage exhibited a higher transmissibility level than its parental XBB.* lineages and is spreading at a rate similar to or slightly faster than the JN.1* lineages.

**IMPORTANCE**  This study highlights the emergence and rapid dissemination of the recombinant SARS-CoV-2 XDR lineage, derived from the Omicron lineages JD.1.1 and JN.1.1. The XDR lineage exhibited equivalent transmissibility to its JN.1* parental lineages and quickly spread across Brazil in late 2023. The findings underscore the critical role of real-time genomic surveillance in detecting novel variants with higher transmission potential. By utilizing phylogenetic and epidemiological methods, this research provides important insights into the molecular dynamics of XDR, which could inform public health responses and vaccine composition updates. The study's significance lies in its ability to document the impact of recombination on viral evolution, offering valuable information to the field of virology and pandemic preparedness.

**KEYWORDS**  SARS-CoV-2, XDR, Brazil, recombination, phylogeography

**Peer Reviewers** Sivasankaran Munusamy Ponnan, Indian Institute of Science Bangalore, Chennai, India; Sanjay Kumar Dey, University of Delhi, Delhi, India

Address correspondence to Gonzalo Bello, gbello@ioc.fiocruz.br.

The authors declare no conflict of interest.

See the funding table on p. 10.

Recombination is a major driving force of virus evolution and has contributed significantly to the genetic diversity of SARS-CoV-2 lineages (1). The first documented example of a SARS-CoV-2 variant increasing its fitness through recombination, rather than substitutions, was the Omicron subvariant XBB that emerged through the recombination of two co-circulating BA.2 lineages, BJ.1 and BM.1.1.1, around mid-2022 (2). The XBB lineage showed a significant capacity to evade population immunity and a substantially higher effective reproduction number ($R_e$) than the parental lineages, suggesting that the recombination event increased viral fitness (2). In early 2023, multiple XBB descendent lineages, such as XBB.1.5*, XBB.1.9.2*, and XBB.1.16*, spread worldwide and rapidly became predominant (3). As a result, in May 2023, the World Health Organization (WHO) Technical Advisory Group on COVID-19 Vaccine Composition recommended using a monovalent XBB.1 descendent lineage, such as XBB.1.5, as the vaccine antigen (4). The upsurge of the XBB lineages highlights the importance of real-time analyses of viral genomes to monitor the potential emergence of novel SARS-CoV-2 recombinant lineages with higher transmissibility, virulence, and/or immune escape properties to guide vaccine composition updates.

In August 2023, a highly mutated Omicron subvariant named BA.2.86, with over 30 mutations in the spike (S) protein concerning BA.2 and XBB subvariants, was identified and classified by the WHO as a variant under monitoring (5). This novel subvariant probably evolved through a saltation-like process directly from a BA.2 ancestor circulating in South Africa around May 2023 (6). The average $R_e$ of BA.2.86 was estimated to be about 1.29-fold greater than that of XBB.1.5 and 1.07 higher than that of EG.5.1, suggesting that BA.2.86 potentially has remarkable fitness considering the XBB variants (7, 8). The BA.2.86 subvariant, however, was not as immune evasive as the XBB lineages circulating in mid-2023 and failed to become dominant at a global scale (6, 8–14). Instead, the BA.2.86 sub-lineage JN.1, with just one additional S mutation (S:L455S), rapidly surged on a global scale in late 2023 and was designated by the WHO as a variant of interest (VOI) in December 2023 (15, 16). JN.1 displayed lower affinity to angiotensin-converting enzyme 2 (ACE2) but enhanced immune escape compared with its predecessor BA.2.86 (Kaku et al., 2024) (11, 17), which partly explains the higher $R_e$ of JN.1 with respect to (w.r.t.) that of BA.2.86.1 and XBB lineages co-circulating in several European countries at the end of November 2023 (18).

The SARS-CoV-2 epidemic in Brazil in 2023 was driven by the spread of different XBB lineages that locally evolved by the stepwise accumulation of mutations in the S protein, which increased the affinity to the ACE2 receptor or the immune escape (19). Genomic data compiled by the COVID-19 Fiocruz Genomic Surveillance Network (https://www.genomahcov.fiocruz.br/dashboard-en/) indicate that XBB lineages dominated the Brazilian epidemic until October 2023 but have been progressively outcompeted by the VOI JN.1* since November 2023. Interestingly, that data also revealed the concomitant spread of a new Omicron subvariant in Brazil, XDR, that arose through recombination between XBB and BA.2.86 co-circulating subvariants. In this study, we explored the origin, dissemination pattern, and transmissibility of this novel XDR recombinant lineage between November 2023 and February 2024.

## MATERIALS AND METHODS

### SARS-CoV-2 Brazilian genome sequences

A total of 5,687 SARS-CoV-2 complete genome sequences recovered across 21 Brazilian states between 1 October 2023 and 30 April 2024 were newly generated by the COVID-19 Fiocruz Genomic Surveillance Network (https://www.genomahcov.fiocruz.br/en/). There was no intended bias considering dense sampling of specific outbreaks, and we do not consider the health status or epidemiological data of individuals as a criterion for sampling. SARS-CoV-2 genome sequences were generated using the COVIDSeq Test Kit (Illumina) as previously described (20), and consensus sequences were produced with the DRAGEN COVID Lineage v4 or ViralFlow 1.0 (21).

All genomes were uploaded to the EpiCoV database of GISAID (https://gisaid.org/). Additionally, all SARS-CoV-2 complete genomes (>29,000 nucleotides) collected in the country between 1 October 2023 and 30 April 2023 and submitted to the EpiCov database until 31 August 2024 with complete collection date and limited presence of unidentified positions ($N$ <5%) were downloaded ($n$ = 4,986). Sequences were classified using the "Phylogenetic Assignment of Named Global Outbreak Lineages" software v4.3.1 (pangolin-data version 1.21) (22, 23).

## XDR data set and maximum likelihood analysis

To analyze the spatiotemporal dynamics of the initial emergence and expansion of SARS-CoV-2 XDR lineage, we compiled a data set comprising all Brazilian ($n$ = 161) and non-Brazilian ($n$ = 25) sequences attributed to this lineage in the first 4 months of its dissemination, collected between 8 November 2023 and 29 February 2023, which were submitted to the EpiCov database until 29 February with the previously described parameters. The resulting data set [29,421 nucleotides from position 1 of ORF1ab to position 117 of ORF10 (24)] was aligned using MAFFT v7.467 (25). To identify the XBB and BA.2.86 sub-lineages that serve as parental to XDR, we created a consensus from the complete XDR data set using Seaview software v.5.0.5 (26) with a 90% identity threshold. From this consensus, the initial 11,043 positions and the final 17,695 positions were used in two local blastn (27) searches against reference data sets comprising all XBB.1.5* ($n$ = 18,303) and JN.1* ($n$ = 1,898) genomes from the EpiCov database sampled in the 2 months prior to the first detection of XDR on 8 November 2024. We selected the top 500 hits from each search, aligned them to their corresponding subgenomic XDR sequence, and conducted a maximum likelihood (ML) phylogenetic analysis using IQ-TREE v2.2.2.7 (28) under the best nucleotide substitution model as selected by the ModelFinder application (29). The approximate likelihood-ratio test (30) assessed the branch support based on the Shimodaira-Hasegawa-like procedure with 1,000 replicates. An ML analysis was also performed for the entire XDR data set using the same parameters.

## Spatiotemporal dynamics of the XDR lineage

The temporal signal of the ML tree of the XDR complete data set was assessed by performing a regression analysis of the root-to-tip divergence against sampling time using TempEst v1.5.36 (31). Sequence outliers were removed from subsequent analyses. The significance of the association between the two variables was assessed with a Spearman correlation test implemented in the R programming language v.4.1.2 (32). A time-scaled Bayesian phylogeographic tree was estimated in BEAST v1.10.4 (33, 34) with BEAGLE (35), under a relaxed uncorrelated molecular clock model (36) with a uniform distribution between 5.0E−4 and 1.5E−3 (37–39), the Bayesian skyline coalescent model (40), and a reversible discrete phylogeographic model (41). The Brazilian sequences were categorized based on their regions of origin ($n$ = 5), while foreign ones were grouped according to their subcontinental regions ($n$ = 3). Markov chain Monte Carlo (MCMC) simulations were run sufficiently long to ensure convergence [effective sample size (ESS) >200] in all parameters as assessed in TRACER v1.7 (42). The number and directionality of location transitions in XDR history were estimated with a Markov Jumps count (43). The maximum clade credibility trees were summarized with TreeAnnotator v1.10 and visualized using Treeio v3.1.7 (44) and ggtree v3.2.1 R packages (45). Graphs used to present the results were generated using the ggplot2 R package (46).

## Effective reproductive number ($R_e$) estimation

The temporal trajectory of the $R_e$ of the XDR lineage in Brazil was estimated from genomic data by using the birth-death skyline (BDSKY) model (47) implemented within BEAST 2 v.2.6.2 (48). The sampling rate was set to zero for the period before the oldest sample and estimated from the data afterward. The BDSKY prior settings were as follows: become uninfectious rate (exponential, mean = 36); reproductive number (log-normal,

mean = 0.8, s.d. = 0.5); sampling proportion (beta, alpha = 1, beta = 275). The origin parameter was conditioned to root height, and $R_e$ was estimated in a piecewise manner over four time intervals defined from the date of the most recent sample up to the root of the tree. A normal prior was applied to the $T_{MRCA}$ of the lineage. The MCMC chain was run until all relevant parameters reached ESS >200, as previously explained. The inferred temporal evolution of the $R_e$ was plotted with the bdskytool R package (https://github.com/laduplessis/bdskytools).

### Relative instantaneous reproduction number ($R_{RI}$) estimations

We measured the transmission advantage of JN* and XDR lineages compared to the XBB* lineages co-circulating in Brazil by estimating the $R_{RI}$ from the observed frequencies of variants in the country from November 2023 to January 2024 as described elsewhere (49). The probability mass function of the generation time of Omicron was modeled by discretizing the gamma distribution with α = 4.03 and θ = 0.737, so that the gamma distribution has the same mean (2.97) and variance (2.19) as the log-normal distribution estimated by Park et al. (50). SARS-CoV-2 lineages observed in each country region were counted using half-month bins. Sequences from Ceará and Bahia were excluded as these states displayed a singular epidemiological molecular pattern within Brazil characterized by very early spread of JN.1* lineages and a high prevalence of the JN.1.23 lineage, respectively (data not shown). By maximizing the likelihood function of the multinomial distribution, we estimated the $R_{RI}$ of JN* and XDR lineages with respect to the XBB* lineages and initial frequencies of variants on 1 November 2023. The trajectories of variant frequencies were calculated by using the maximum likelihood estimates of the parameters in the model, and the 95% confidence interval (CI) of the trajectories was calculated by using combinations of parameters within the 95% confidence region (51).

### Data on hospitalizations for severe acute respiratory illness

We extracted data about hospitalizations resulting from severe acute respiratory illness (SARI) attributed explicitly to SARS-CoV-2 (SARI-COVID) in Brazil from November 2023 to January 2024. This information was sourced from the Influenza Surveillance Information System (SIVEP-Gripe) database (https://opendatasus.saude.gov.br/ data set?tags = SRAG), as previously described (19).

## RESULTS AND DISCUSSION

Among SARS-CoV-2 nearly complete genome sequences submitted to the EpiCoV database of GISAID until 31 August 2024, we identified 695 sequences classified as XDR between 8 November 2023, and 30 April 2024. Most of the XDR sequences were sampled in Brazil (86%), including the earliest sequences sampled. The remaining XDR sequences were sampled in neighboring South American countries (Argentina, Chile, Paraguay, Peru, and Uruguay, 2%), North America (Canada and USA, 5%), and Europe (Belgium, Finland, Denmark, Spain, Sweden, and United Kingdom, 3%). The XDR recombinant genomes contain all the JD.1.1 and none of the JN.1.1 lineage-defining mutations until the mutation ORF1a:V3593F, and all JN.1.1 lineage-defining mutations starting from the mutation ORF1a:R3821K, thus supporting a recombination breakpoint between nucleotide positions 11043 and 11726 of the ORF1a coding region (Fig. 1a). In order to confirm the parental lineages of the XDR recombinant, we performed a BLAST search analysis to detect the most similar sequences to the 5′ and 3′ ends flanking the recombination breakpoint among XBB.1.5* and JN.1* sequences globally sampled over 2 months before the detection of the first XDR genome and that were submitted to the GISAID database until 5 March 2024. The ML phylogenetic analysis supports that the most closely related sequences to the XDR recombinant's 5′ and 3′ ends correspond to JD.1.1 and JN.1.1 lineages, respectively (Fig. S1a and b).

To understand the position of the XDR lineage within the evolving molecular landscape of the SARS-CoV-2 epidemic in Brazil, we analyzed 10,673 Brazilian

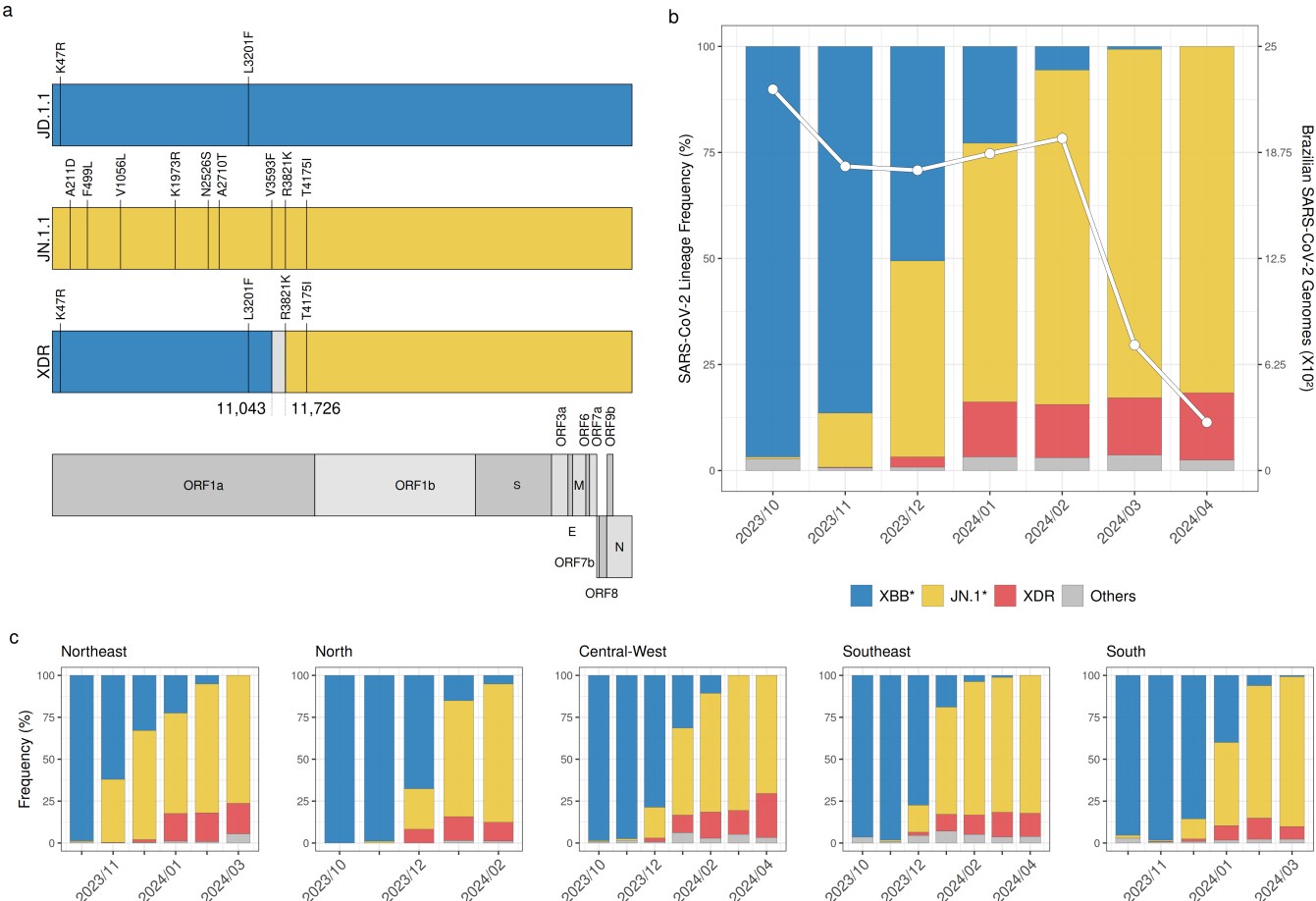

**FIG 1** Genomic structure and prevalence of SARS-CoV-2 XDR lineage in Brazil. (a) Recombinant structure of the SARS-CoV-2 XDR lineage. Synapomorphic amino acid mutations of the XDR lineage and the most probable parental lineages (JD.1.1 and JN.1.1) compared to the BA.2 lineage are highlighted. The genomic coordinates within ORF1a that flank the recombination point are also annotated. Each segment of the XDR genome is color coded based on its parental lineage, with the segment that potentially houses the recombination point indicated in light gray. (b) Relative prevalence of the XDR lineage in Brazil's SARS-CoV-2 epidemic between October 2023 and February 2024 is shown alongside the monthly count of viral genomes sampled in the country ($n_{TOTAL}$ = 6,152). (c) Relative prevalence of the XDR lineage stratified by the country's regions using the same color code provided in the previous panel. The main lineage groups considered (XBB*, JN.1.*, and XDR) are colored according to the legend in the previous panel. In panels a–c, only months with more than 50 genomes represented are shown.

SARS-CoV-2 genomes collected between 1 October 2023 and 29 April 2024 (Fig. 1b). As of October 2023, most SARS-CoV-2 sequences in Brazil belong to the XBB* lineages (Fig. 1b). The first JN.1* lineages were identified in October 2023 across the Central-Western and Southern regions, and their prevalence steadily rose from <1% in that month to 60% in February 2024 (Fig. 1b). Concurrently, the first XDR sequences were detected in November 2023 in the Northeastern region, and their prevalence surged from <1% in that month to nearly 15% by January 2024, marking a significant increase in the frequency of this recombinant lineage in the study period (Fig. 1b). The relative prevalence of XDR remained stable between January and April 2024 despite the progressive extinction of the XBB* lineages and the concomitant dominance of the JN.1* lineages in the country's genomic profile during that time (Fig. 1b). The same overall pattern of substitution of lineages XBB by JN.1* and XDR was consistently observed in the whole country since October 2023 (Fig. 1c). The expansion of the JN.1.* and XDR lineages started in the Northeastern region and culminated in the Southern region, and by March 2023, the XBB* lineages had nearly disappeared from all country regions.

We conducted a Bayesian phylogeographic inference to ascertain the spatiotemporal dissemination pattern of the XDR lineage. After removing low-quality sequences with

high frequency of private mutations or missing data of the entire receptor-binding domain (RBD) of the S protein, we retained a total of 186 XDR sequences from Brazilian Northeastern region (41%), Brazilian Southeastern region (9%), Brazilian Central-Western region (22%), Brazilian Northern region (13%), Brazilian Southern region (2%), other South American countries (7%), North America (4%), and Europe (2%). The regression analysis of the root-to-tip divergence against sampling time revealed a satisfactory temporal signal in the XDR data set (Fig. 2a), and the Bayesian time-scaled phylogenetic analysis estimated an evolutionary rate of $7.5 \times 10^{-4}$ (95% HPD (highest posterior density): $5.1–9.7 \times 10^{-4}$) substitutions/site/year, consistent with that inferred for other SARS-Cov-2 lineages (37–39). The Bayesian phylogeographic analysis indicates that the XDR lineage likely arose in the Northeastern region ($PSP = 0.99$) around late October 2023 (Fig. 2b). From mid-November 2023 onward, the XDR lineage spread mainly from the Northeastern to other Brazilian regions and beyond Brazil's borders (Fig. 2c and d).

We then applied the BDSKY model to estimate the $R_e$ of the XDR lineage by selecting all sequences from Brazil. The temporal trajectory of the median $R_e$ supports an initial expansion phase ($R_e$ ~1.5) during November and December 2023, followed by a phase of stabilization ($R_e$ ~1) in January 2024 and subsequent decrease ($R_e$ <1) in February 2024 (Fig. 3a; Table S1). Interestingly, the trajectory of the $R_e$ of the XDR lineage coincides with the overall pattern of severe acute respiratory illness cases detected in the Northeastern Brazilian region during that period (Fig. 3a). We speculate that the recent dissemination of the XDR lineage outside the epicenter was probably not captured by our BDSKY model

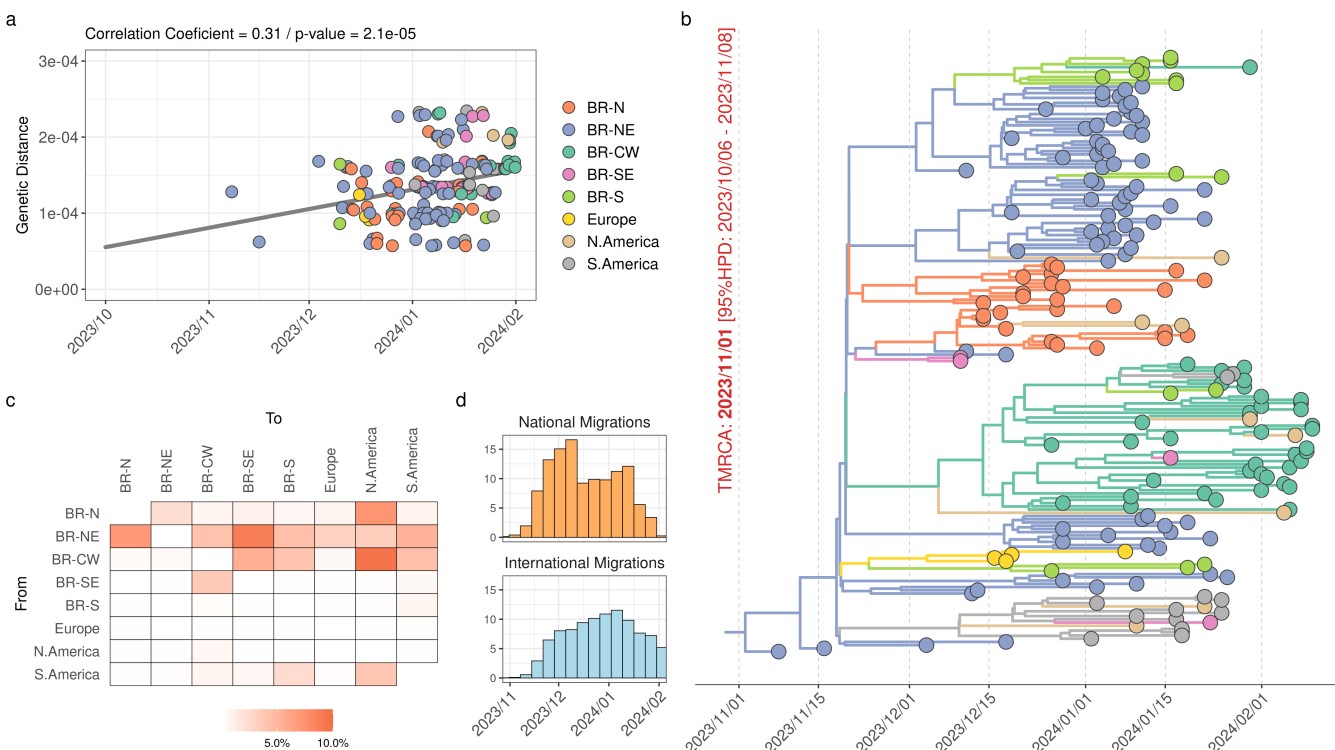

**FIG 2** Spatiotemporal dynamics of global dissemination of SARS-CoV-2 XDR lineage. (a) Plot of the root-to-tip divergence against collection dates of XDR genomes sampled worldwide ($n = 186$) between November 2023 and February 2024. The correlation coefficient and *P*-value obtained in the analysis are annotated at the top of the panel. Data points are colored according to the sampling location, as indicated in the legend. (b) Time-scale maximum clade credibility Bayesian tree of the XDR lineage. Tip and branch colors indicate the sampling state and the most probable inferred state of the nodes, respectively, as indicated in the legend. The inferred $T_{MRCA}$ of the lineage is annotated near the root node of the tree. All horizontal branch lengths are time scaled, and the tree was automatically rooted under the assumption of the molecular clock model. (c) Heatmap cells are colored according to the estimated number of viral migrations between locations based on the Markov Jump counts in the Bayesian phylogeographic analysis. (d) Estimated number of viral migrations among Brazilian regions (*National Migrations*) and between any Brazilian region and any foreign country (*International Migrations*) through time. BR, Brazil; N, North; NE, Northeast; CW, Central-west; SE, Southeast; S, South.

and that more sequences collected at recent times are probably necessary to accurately estimate the $R_e$ of the XDR lineage at February 2024. The median $R_e$ of the XDR lineage in late 2023 was lower than those estimated for different SARS-CoV-2 lineages spreading in distinct Brazilian regions during epidemic waves in 2020–2022, such as the B.1* ($R_e$ = 2.0–3.8), Gamma P.1 ($R_e$ = 2.0–2.6), and Omicron BA.1 ($R_e$ = 2.5–3.0) (20, 52, 53). This supports that current Brazilian population immunity probably confers some level of protection against reinfections by XDR (and also probably JN.1*) lineages.

To trace the shift in viral fitness associated with the emergence of the XDR recombinant, we estimated the $R_{RI}$ of XDR and JN* (BA.2.86 + S:L455S) lineages w.r.t. XBB* lineages co-circulating in Brazil from the observed frequencies of variants in the country from November 2023 to January 2024. These analyses support that JN* and XDR lineages were more transmissible than XBB* lineages. The average $R_{RI}$ of JN* and XDR lineages in Brazil was estimated to be 1.29 and 1.37 times higher than that of XBB* co-circulating lineages (mainly JD.1* and GK.1*), respectively (Fig. 3b). Thus, the average $R_{RI}$ of XDR lineage was 1.06 times higher than that of JN* lineages. We also obtained specific $R_{RI}$ values for the Northeastern, Southeastern, and Central-Western Brazilian regions. All analyses point to a higher transmissibility of the XDR lineage w.r.t. the XBB* lineages across all country regions and w.r.t. the JN* lineages only in the Northeastern region (Fig. 3c). Interestingly, the average $R_{RI}$ of JN* and XDR lineages w.r.t. XBB lineages co-circulating in Brazil in late 2023 was quite similar to the average $R_{RI}$ of XBB* lineages w.r.t. BA.5 lineages (mainly BQ.1*/BE*) co-circulating in Brazil in early 2023 ($R_{RI}$ = 1.24–1.48) (19) and was also similar to the estimated relative $R_e$ of JN.1 w.r.t. XBB lineages co-circulating in Europe such as EG.5.1, HK.3, and JD.1* (~1.10–1.40) (18).

The inference that the viral transmissibility of the XDR lineage was comparable to its parental JN* lineage, yet significantly higher than its other parental XBB* lineage, is supported by the stable persistence of XDR amid the progressive decline of the XBB*

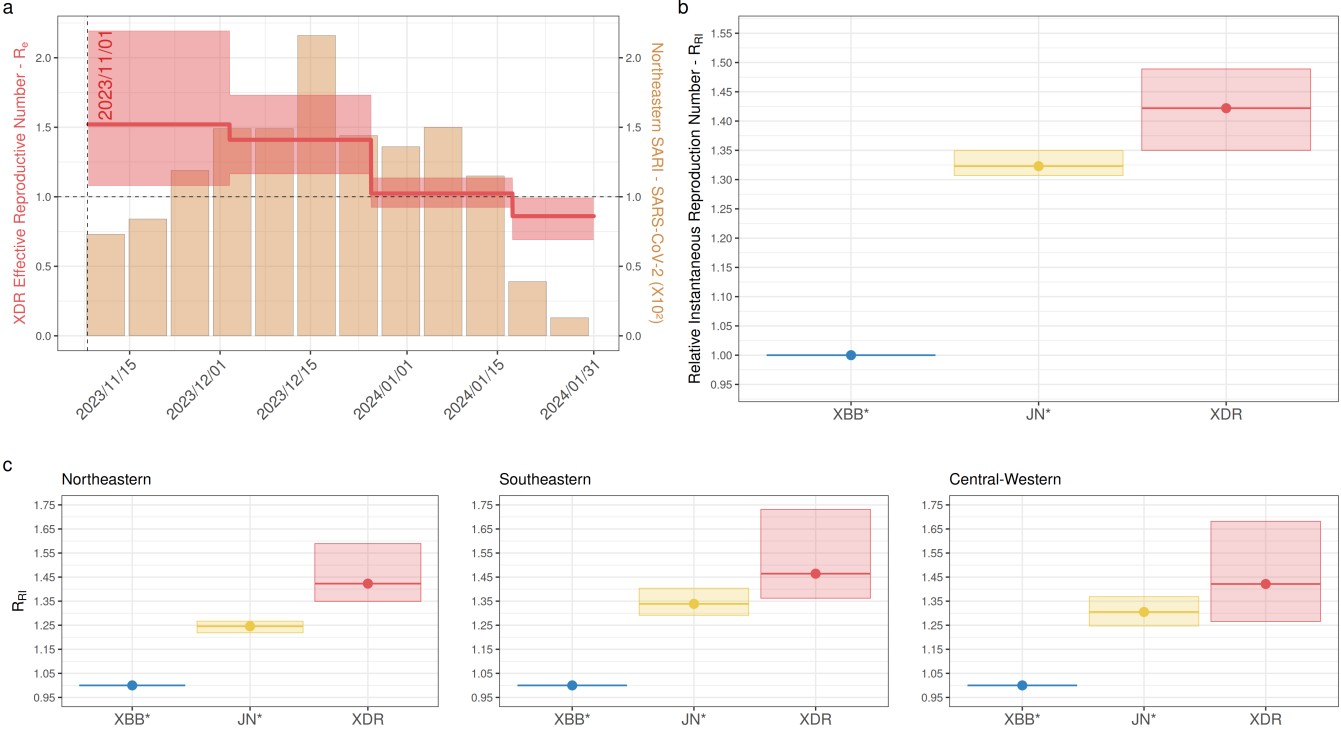

**FIG 3** Absolute and relative $R_e$ of SARS-CoV-2 XDR lineage in Brazil. (a) Temporal variation in the $R_e$ of the XDR lineage circulating in Brazil was estimated using the BDSKY approach, juxtaposed with the monthly count of SARI cases attributed to SARS-CoV-2 in the Brazilian Northeastern region. The median (solid red line) and the 95% HPD interval (shaded red area) of the $R_e$ estimates are represented together with the estimated $T_{MRCA}$ of the lineage (vertical shaded line). (b and c) Relative transmissibility ($R_{RI}$) of XDR and JN.* lineages w.r.t. the XBB.* lineages estimated from the observed frequencies of variants circulating in Brazil (b) or in specific country regions (c). The average (solid lines) and the 95% CI interval (shaded area) of the $R_{RI}$ estimates are represented.

lineage in Brazil between January and April 2024. This contrasts with the evolutionary dynamics observed during the emergence of XBB* in 2022, where XBB*, a recombinant of two BA.2* lineages, exhibited viral transmissibility that surpassed both of its parental lineages (54). Importantly, while the XDR lineage inherited the full S protein from its parental JN.1.1 lineage, the XBB strain is a recombinant with a chimeric S protein because the recombination breakpoint is between positions 22897 and 22941, within the receptor-binding domain of the S protein (corresponding to amino acid positions 445–460) (54). Given the pivotal role of the S protein in shaping the SARS-CoV-2 intrinsic transmissibility and immune escape (55), the XDR's transmissibility in relation to its parental lineages can be a consequence of the specific recombination pattern that underpinned its emergence.

Interestingly, the XDR lineage was 1 of 13 designated lineages that emerged from the recombination between XBB* and BA.2.86*/JN.1* lineages until August 2024 (Table S2) (22). These recombinants were first detected between early October 2023 (XDU) and late January 2024 (XEA), coinciding with the period of co-circulation of parental lineages (56). There was no spatial structure associated with the emergence of these lineages as within the first month of XDU's detection in Asia, additional recombinants had already been identified in Europe (XDD and XDQ), North America (XDP), and South America (XDR). Notably, most (~80%) of these recombinants acquired the full S protein from the BA.2.86*/JN.1* parental lineages. A small fraction of recombinants (XDN, XDQ, and XDT) displayed a chimeric S protein resulting from a mixture of BA.2.86*/JN.1* and XBB* parental sequences, but all of them harbor the amino terminal domain (NTD) and the RBD of the BA.2.86*/JN.1* lineages. These findings support an overall transmission advantage for viral variants carrying the complete or the NTD + RBD domains of the S protein of BA.2.86*/JN.1* lineages.

This study has a number of limitations. Notably, we lack experimental evidence to support the increased viral fitness of XDR lineage inferred from genomic data. To provide such evidence, it would be necessary to test the immune escape and intrinsic transmissibility properties (57) of the XDR lineage in comparison to those of the parental lineages. Specifically, to assess immune escape, one would need to demonstrate, for example, a reduction in the PRNT50 titer of XDR isolates against plasma from individuals with XBB* infections, similar to the observations made with the JN.1 lineage (17). Additionally, due to the lack of comprehensive data on the total number of infections in Brazil, we relied on SARI-COVID cases to estimate the real-world impact of XDR emergence. This approach, nonetheless, may fail to capture the broader transmission dynamics of XDR as it excludes asymptomatic and mild cases, which constitute a significant portion of SARS-CoV-2 infections.

In summary, our analyses suggest that the XDR lineage likely originated from a recombination event between SARS-CoV-2 lineages JD.1.1 and JM.1.1 in the Northeastern region of Brazil in late October 2023 and was detected just a couple of weeks after its emergence. The XDR lineage spread to other Brazilian regions and beyond Brazil's borders from the Northeastern region. Our findings suggest that the XDR recombinant lineage exhibited higher transmissibility than XBB* lineages and was comparable to the JN.1* lineages across all Brazilian regions. These findings underscore the importance of real-time molecular surveillance of SARS-CoV-2 cases for rapidly detecting new emergent viral variants with increased transmissibility and potential to spread globally. Moreover, this study emphasizes the relevance of monitoring the emergence of SARS-CoV-2 recombinant lineages between XBB and BA.2.86 Omicron subvariants and investigating potential determinants of viral transmissibility located within and outside the S region.

## ACKNOWLEDGMENTS

We gratefully acknowledge all data contributors, that is, the originating laboratories responsible for obtaining the specimens and the submitting laboratories for generating the genetic sequence and metadata and sharing via EpiCov database from GISAID, on which this research is based. The authors also wish to thank all the healthcare workers

and scientists who have worked hard to deal with this pandemic threat. In addition, we appreciate the support of the Respiratory Viruses Genomic Surveillance Network of the General Laboratory Coordination (CGLab) of the Brazilian Ministry of Health (MoH) and Brazilian Central Laboratory States (LACENs).

This study was supported by the Department of Science and Technology (DECIT) of the Brazilian Ministry of Health (MoH); CGLab/MoH (General Laboratories Coordination of Brazilian Ministry of Health); UK Health Security Agency (UKHSA) by the New Variant Assessment Platform (NVAP) project; the Japan International Cooperation Agency (JICA); CVSLR/FIOCRUZ (Coordination of Health Surveillance and Reference Laboratories of Oswaldo Cruz Foundation); Centers for Disease Control and Prevention (CDC) grant; CNPq COVID-19 (MCTI402457/2020-0 and 403276/2020–9); INOVA Fiocruz (VPPCB-005-FIO-20–2 and VPPCB-007-FIO-18-2-30); FAPERJ (E26/210.196/2020); FAPEAM (Rede Genômica de Vigilância em Saúde-REGESAM); FAPEAM (INICIATIVA AMAZÔNIA + 10 [grant: 01.02.016301.00439/2023–70]); FAPEAM/INOVA FIOCRUZ INOVAÇÃO NA AMAZÔNIA (Chamada Pública no. 04/2022); NPI EXPAND–U.S. Agency for International Development (USAID) implemented by Palladium (7200AA19CA00015), Centers for Disease Control and Prevention (CDC Grant Award 002174), and CNPQ CABBIO (grant number 423857/2021–5); and FAPERJ (grant number E-26/211.125/202). G.L.W. had support from a CNPQ productivity research fellowship (307209/2023–7). P.C.R. had support from a CNPQ productivity research fellowship (311759/2022–0). M.M.S. had support from a CNPq productivity research fellowship (313403/2018–0). F.G.N. had support from a CNPq productivity research fellowship (306146/2017–7). G.B. had support from FAPERJ (grant number E-26/202.896/2018) and CNPq productivity research fellowship (304883/2020–4). I.A. had support from FAPERJ-Fundação Carlos Chagas Filho de Amparo à Pesquisa do Estado do Rio de Janeiro (grant SEI-260003/019669/2022). This research was supported by the FINDINGS Project (Project for the Enhancement of Genomic Monitoring Network for Covid-19) agreed upon between FIOCRUZ (Oswaldo Cruz Foundation), Brazilian Cooperation Agency, and JICA (Japan International Cooperation Agency) on 13 March 2023. This study was partially supported by the Coordenação de Aperfeiçoamento de Pessoal de Nível Superior–CAPES-Finance Code 001.

The study was conceived and designed by G.B., F.G.N., and I.A., F.M., R.K., P.C.R., G.L.W., and F.G.N. contributed to diagnostics and sequencing analysis. K.I. contributed to the instantaneous reproduction number estimations. M.G., F.C.D.C., and W.A.F.D.A. worked on the retrieval and analysis of Brazilian epidemiological data. F.G.N. and M.M.S. contributed to laboratory management and obtaining financial support. The FGSN consortium contributed to the generation of genomic sequences. I.A., K.I., and E.C.P. performed the bioinformatics analysis. I.A. and G.B. wrote the first draft, and all authors contributed and approved the final manuscript.

The representatives of the Fiocruz Genomic Surveillance Network in Brazil are Hazerral Hazerral de Oliveira Santos (LACEN, AL), Ana Flavia Mendonça (LACEN, GO), Gislene Garcia de Castro Lichs (LACEN, MS), Adelino Soares Lima Neto (LACEN, PI), Patricia Brasil (INI, RJ), Andréa Cony (LACEN, RJ), Jayra Juliana Paiva Alves Abrantes (LACEN, RN), Tatiana Schäffer Gregianini (LACEN, RS), Darcita Buerger Rovaris (LACEN, SC), Cliomar Alves dos Santos (LACEN, SE), Franciano Dias Pereira Cardoso (LACEN, TO), Zoraida del Carmen Fernandez Grillo (FIOCRUZ, MS), Adriano Abbud (Instituto Adolfo Lutz, SP), Luana Barbagelata (Instituto Evandro Chagas, PA), Leandro Cavalcante Santos (LACEN, AC), Márcia Socorro Pereira Cavalcante (LACEN, AP), Valnete Andrade (LACEN, PA), Thiago Franco de Oliveira Carneiro (LACEN, PB), Shirlene Telmos Silva de Lima (LACEN, CE), Jórdan Barros da Silva (LACEN, DF), Rodrigo Ribeiro Rodrigues (LACEN, ES), Cliomar Alves dos Santos (LACEN, SE).

## AUTHOR AFFILIATIONS

[1]Laboratório de Arbovírus e Vírus Hemorrágicos, Instituto Oswaldo Cruz, Fiocruz, Rio de Janeiro, Brazil

[2]International Institute for Zoonosis Control, Hokkaido University, Hokkaido, Japan

[3]Grupo de Métodos Analíticos em Vigilância Epidemiológica, Fiocruz, Rio de Janeiro, Brazil

[4]Departamento do Programa Nacional de Imunizações, Coordenação-Geral de Vigilância das doenças imunopreveníveis, Secretaria de Vigilância em saúde e ambiente, Brasília, Brazil

[5]Instituto Gonçalo Moniz, Fiocruz, Salvador, Brazil

[6]Fiocruz Ceará, Fortaleza, Brazil

[7]Department of Arbovirology, Bernhard Nocht Institute for Tropical Medicine, WHO Collaborating Center for Arbovirus and Hemorrhagic Fever Reference and Research, National Reference Center for Tropical Infectious Diseases, Hamburg, Germany

[8]Instituto Aggeu Magalhães, Fundação Oswaldo Cruz, Recife, Pernambuco, Brazil

[9]Núcleo de Vigilância de Vírus Emergentes, Reemergentes ou Negligenciados, Laboratório de Ecologia de Doenças Transmissíveis na Amazônia, Instituto Leônidas e Maria Deane, Fiocruz, Manaus, Brazil

[10]Laboratório de Vírus Respiratórios, Exantemáticos, Enterovírus e Emergências Virais, Instituto Oswaldo Cruz, Fiocruz, Rio de Janeiro, Brazil

## AUTHOR ORCIDs

Ighor Arantes http://orcid.org/0000-0002-4131-5338
Fabio Miyajima http://orcid.org/0000-0002-1347-4825
Gabriel Luz Wallau http://orcid.org/0000-0002-1419-5713
Gonzalo Bello http://orcid.org/0000-0002-2724-2793

## FUNDING

| Funder | Grant(s) | Author(s) |
| --- | --- | --- |
| Conselho Nacional de Desenvolvimento Científico e Tecnológico (CNPq) | 311759/2022-0 | Paola Cristina Resende |
| Conselho Nacional de Desenvolvimento Científico e Tecnológico (CNPq) | 313403/2018-0 | Marilda Mendonça Siqueira |
| Conselho Nacional de Desenvolvimento Científico e Tecnológico (CNPq) | 306146/2017-7 | Felipe Gomes Naveca |
| Fundação Carlos Chagas Filho de Amparo à Pesquisa do Estado do Rio de Janeiro (FAPERJ) | E-26/202.896/2018 | Gonzalo Bello |
| Conselho Nacional de Desenvolvimento Científico e Tecnológico (CNPq) | 304883/2020-4 | Gonzalo Bello |
| Fundação Carlos Chagas Filho de Amparo à Pesquisa do Estado do Rio de Janeiro (FAPERJ) | SEI-260003/019669/2022 | Ighor Arantes |
| Conselho Nacional de Desenvolvimento Científico e Tecnológico (CNPq) | 307209/2023-7 | Gabriel Luz Wallau |
| Conselho Nacional de Desenvolvimento Científico e Tecnológico (CNPq) | 423857/2021-5 | Paola Cristina Resende |
| Fundação de Amparo à Pesquisa do Estado do Amazonas (FAPEAM) | 01.02.016301.00439/2023-70 | Felipe Gomes Naveca |

## AUTHOR CONTRIBUTIONS

Ighor Arantes, Conceptualization, Data curation, Formal analysis, Investigation, Methodology, Validation, Visualization, Writing – original draft, Writing – review and editing | Kimihito Ito, Formal analysis, Methodology, Writing – review and editing | Marcelo Gomes, Data curation, Formal analysis, Writing – review and editing | Felipe Cotrim de Carvalho, Data curation, Writing – review and editing | Walquiria Aparecida

Ferreira de Almeida, Data curation, Writing – review and editing | Ricardo Khouri, Data curation, Funding acquisition, Writing – review and editing | Fabio Miyajima, Data curation, Funding acquisition, Writing – review and editing | Gabriel Luz Wallau, Data curation, Funding acquisition, Writing – review and editing | Felipe Gomes Naveca, Data curation, Funding acquisition, Writing – review and editing | Elisa Cavalcante Pereira, Formal analysis, Methodology, Writing – review and editing | Marilda Mendonça Siqueira, Data curation, Funding acquisition, Writing – review and editing | Paola Cristina Resende, Data curation, Funding acquisition, Writing – review and editing | Gonzalo Bello, Conceptualization, Data curation, Methodology, Supervision, Writing – original draft, Writing – review and editing.

## DATA AVAILABILITY

This study's conclusions derive from examining 5,687 SARS-CoV-2 genomes from Brazil, which have been made publicly accessible via the EpiCov database from GISAID. These genomes were collected after 1 October 2023, and submissions were recorded up until 30 April 2024. The data can be accessed at https://doi.org/10.55876/gis8.240904tw. For our investigation of the XDR parental lineages and the phylogeographic analysis, we further included 25,187 global reference sequences collected after 8 September 2023, and submissions were also recorded up until 30 April 2024, which are available at https://doi.org/10.55876/gis8.240904ko. The XML files utilized throughout this analysis are openly accessible for review and replication of our methods at https://github.com/larboh-ioc/sars2_xdr.

## ETHICS APPROVAL

This study was approved by the Ethics Committee of the FIOCRUZ (CAAE: 68118417.6.0000.5248 and CAAE:32333120.4.0000.5190), which waived signed informed consent for all participants. All methods followed guidelines and regulations of the Brazilian Ministry of Health.

## ADDITIONAL FILES

The following material is available online.

### Supplemental Material

**Supplemental figure and tables (Spectrum01193-24-s0001.docx).** Fig. S1; Tables S1 and S2.

### Open Peer Review

**PEER REVIEW HISTORY (review-history.pdf).** An accounting of the reviewer comments and feedback.

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
