## [Reviewer comments · Microbiology Spectrum]

Microbiology Spectrum

Rapid spread of the SARS-CoV-2 Omicron XDR lineage derived from recombination between XBB and BA.2.86 subvariants circulating in Brazil in late 2023

Ighor Arantes, Kimihito Ito, Marcelo Gomes, Felipe Carvalho, Walquiria de Almeida, Ricardo Khouri, Fabio Miyajima, Gabriel Wallau, Felipe Naveca, Elisa Pereira, Marilda Siqueira, Paola Resende, and Gonzalo Bello

Corresponding Author(s): Ighor Arantes, Fundacao Oswaldo Cruz

Review Timeline:

Submission Date:	May 21, 2024
Editorial Decision:	August 27, 2024
Revision Received:	September 10, 2024
Accepted:	September 21, 2024

Editor: Ujjwal Neogi

Reviewer(s): Disclosure of reviewer identity is with reference to reviewer comments included in decision letter(s). The following individuals involved in review of your submission have agreed to reveal their identity: Sivasankaran Munusamy Ponnar (Reviewer #1); Sanjay Kumar Dey (Reviewer #2)

Transaction Report:

DOI: <https://doi.org/10.1128/spectrum.01193-24>

Re: Spectrum01193-24 (Rapid spread of the SARS-CoV-2 Omicron XDR lineage derived from recombination between XBB and BA.2.86 subvariants circulating in Brazil in late 2023)

Dear Dr. Ighor Arantes:

Thank you for the privilege of reviewing your work. Below you will find my comments, instructions from the Spectrum editorial office, and the reviewer comments.

Revision Guidelines

Sincerely,
Ujjwal Neogi
Editor
Microbiology Spectrum

Reviewer #1 (Comments for the Author):

The manuscript represents a very interesting phylogenetic analysis of XDR variants of SARS-CoV-2 in Brazil. The authors performed an analysis of the mutations in the new variants of SARS-CoV-2 and found that XDR lineage displayed a recombination breakpoint in the ORF1a coding region, and the most closely related sequences to the 5' and 3' ends of the recombinant correspond to JD.1.1 and JN.1.1 lineages. Interestingly, they found that the XDR lineage exhibited a higher

transmissibility rate compared with its parental XBB lineages and similar or slightly faster than the JN.1 lineage. The author showed the dissemination pattern and transmissibility of this novel XDR recombinant lineage between November 2 and February 2024

Many papers from different regions of the world reported that the XBB variant was found to be able to evade immunity gained through available COVID-19 vaccines and even previous infection with SARS-CoV-2. Moreover, the XBB variant was found to be more transmissible and able to emerge as the next dominant variant and suppress the currently dominant SARS-CoV-2 variants. The paper presents an intriguing discovery of a new Omicron subvariant in Brazil named XDR, which emerged through recombination between the XBB and BA.2.86 co-circulating subvariants.

I have the following minor comments to improve the quality of the manuscript.

- The introduction is well-written. However, the author could add some more background about the XDR lineage, how these variants evade the immune system, and their transmissibility rate etc.,
- It is recommended to revisit the method section, and it can be shortened.
- Did the author collect any immunological data from those participants?
- Figures: Requested author to change the colour for "years" and other information showed in light grey. It would be very eye-catching to the reader if a change in different colour.
- Figures: Need higher DPI figures.
- Although the article discusses the time period between November 2 and February 2024, I am curious about the current status. Are there any follow-up studies?

Reviewer #2 (Comments for the Author):

The authors have done a commendable job by highlighting the risk and reason for the rapid spread of new variants of SARS-CoV-2, at least for the Brazilian population. The following queries may be addressed to improve it further:

- Can it be extrapolated as a common mechanism for the rapid spread of the new variants in other countries and/or continents? If yes, explain/highlight the same in the discussion with proper logic.
- Is the same mechanism true for upcoming mutations as well? If not, what is the significance of the current study if these mutations/variants are lost by some time?
- What can be an experimental proof than just statistical analysis for your study conclusion?

The manuscript represents a very interesting phylogenetic analysis of XDR variants of SARS-CoV-2 in Brazil. The authors performed an analysis of the mutations in the new variants of SARS-CoV-2 and found that XDR lineage displayed a recombination breakpoint in the ORF1a coding region, and the most closely related sequences to the 5' and 3' ends of the recombinant correspond to JD.1.1 and JN.1.1 lineages. Interestingly, they found that the XDR lineage exhibited a higher transmissibility rate compared with its parental XBB lineages and similar or slightly faster than the JN.1 lineage. The author showed the dissemination pattern and transmissibility of this novel XDR recombinant lineage between November 2 and February 2024

Many papers from different regions of the world reported that the XBB variant was found to be able to evade immunity gained through available COVID-19 vaccines and even previous infection with SARS-CoV-2. Moreover, the XBB variant was found to be more transmissible and able to emerge as the next dominant variant and suppress the currently dominant SARS-CoV-2 variants. The paper presents an intriguing discovery of a new Omicron subvariant in Brazil named XDR, which emerged through recombination between the XBB and BA.2.86 co-circulating subvariants.

I have the following minor comments to improve the quality of the manuscript.

- The introduction is well-written. However, the author could add some more background about the XDR lineage, how these variants evade the immune system, and their transmissibility rate etc.,
- It is recommended to revisit the method section, and it can be shortened.
- Did the author collect any immunological data from those participants?
- Figures: Requested author to change the colour for "years" and other information showed in light grey. It would be very eye-catching to the reader if a change in different colour.
- Figures: Need higher DPI figures.
- Although the article discusses the time period between November 2 and February 2024, I am curious about the current status. Are there any follow-up studies?

Reviewer #1 (Comments for the Author):

The manuscript represents a very interesting phylogenetic analysis of XDR variants of SARS-CoV-2 in Brazil. The authors performed an analysis of the mutations in the new variants of SARS-CoV-2 and found that XDR lineage displayed a recombination breakpoint in the ORF1a coding region, and the most closely related sequences to the 5' and 3' ends of the recombinant correspond to JD.1.1 and JN.1.1 lineages. Interestingly, they found that the XDR lineage exhibited a higher transmissibility rate compared with its parental XBB lineages and similar or slightly faster than the JN.1 lineage. The author showed the dissemination pattern and transmissibility of this novel XDR recombinant lineage between November 2 and February 2024

Many papers from different regions of the world reported that the XBB variant was found to be able to evade immunity gained through available COVID-19 vaccines and even previous infection with SARS-CoV-2. Moreover, the XBB variant was found to be more transmissible and able to emerge as the next dominant variant and suppress the currently dominant SARS-CoV-2 variants. The paper presents an intriguing discovery of a new Omicron subvariant in Brazil named XDR, which emerged through recombination between the XBB and BA.2.86 co-circulating subvariants.

I have the following minor comments to improve the quality of the manuscript.

1.1) *The introduction is well-written. However, the author could add some more background about the XDR lineage, how these variants evade the immune system, and their transmissibility rate etc.,*

Answer: To our knowledge, this is the first and unique work about the XDR lineage published so far. Thus, there is no information about the intrinsic transmissibility or evasion immune properties of this recombinant variant.

1.2) *It is recommended to revisit the method section, and it can be shortened.*

Answer: Following the reviewer's comment, we reduce the method section by ~10%, going from 1,292 to 1,179 words.

1.3) *Did the author collect any immunological data from those participants?*

Answer: Unfortunately, we have no information about immunological data or history of previous infections and vaccine doses in the studied population.

1.4) *Figures: Requested author to change the colour for "years" and other information showed in light grey. It would be very eye-catching to the reader if a change in different colour.*

Answer: Following the reviewers' comments, the information previously displayed in light gray in the manuscript's figures has been updated to more vivid colors.

1.5) Figures: Need higher DPI figures.

Answer: The original submission did have figures with a low DPI; however, this has been corrected in the current version of the manuscript.

1.6) Although the article discusses the time period between November 2 and February 2024, I am curious about the current status. Are there any follow-up studies?

Answer: The original version of the manuscript reported XDR prevalence in the country from November 2023 to February 2024, both at the national and regional levels. In the current version, this time window has been extended to include March and April 2024 as well, with the exception of the Northern region that extends to March due to the very low number of genomes from April (<30). The number of Brazilian genomes available since May is insufficient to confidently infer the genomic profile in the country. These results are presented in Figure 1, panels b and c. The new data confirms a stable XDR prevalence of approximately 15% during this period at the national level.

Reviewer #2 (Comments for the Author):

The authors have done a commendable job by highlighting the risk and reason for the rapid spread of new variants of SARS-CoV-2, at least for the Brazilian population. The following queries may be addressed to improve it further:

2.1) Can it be extrapolated as a common mechanism for the rapid spread of the new variants in other countries and/or continents? If yes, explain/highlight the same in the discussion with proper logic. The same dynamics of recombination and immune escape that led to the rapid spread of the XDR lineage in Brazil could also apply to other regions where similar variants might arise. For instance, the global spread of the Omicron XBB lineage in 2023 demonstrated that recombination events can produce highly transmissible variants capable of evading immune responses, leading to widespread dissemination across continents.

Answer: The rapid spread of the SARS-CoV-2 Omicron XDR lineage in Brazil can be extrapolated as a common mechanism for the swift dissemination of new variants in other countries and continents. It is important to note, however, that recombination is not the only dynamic driving SARS-CoV-2 evolution; the process is also influenced by factors such as mutations and migration events. When comparing the recombination event that led to the emergence of XBB* variants with that of XDR, there is a crucial difference. In the case of XDR,

the recombination breakpoint occurred outside the Spike, resulting in a lineage whose immune escape properties are probably equivalent to that of the JN.1 parental lineage. The XBB* variant, on the other hand, arose from a recombination event with a breakpoint within the Spike protein, producing a protein distinct from both parental lineages. As a result, the XBB* lineage exhibited superior immune escape when compared to both parental lineages (Tamura et al., 2023). Therefore, while recombination events can be expected to occur with relative high frequency, their consequences are not always predictable. We briefly discuss this point in the new version of the manuscript.

2.2) *Is the same mechanism true for upcoming mutations as well? If not, what is the significance of the current study if these mutations/variants are lost by some time?*

Answer: The question of whether the same mechanism applies to upcoming mutations as well depends on the specific evolutionary pressures and circumstances surrounding those mutations. For instance, in the context of the SARS-CoV-2 Omicron XDR lineage, which is a recombinant variant derived from the XBB and BA.2.86 subvariants, the mechanism of recombination played a crucial role in its emergence and rapid spread in Brazil during late 2023. However, this mechanism might not be universally applicable to all future variants. Some mutations might arise through different processes, such as stepwise accumulation of point mutations or selective pressures like evasion of established immunity. Each mechanism could result in variants with distinct transmissibility, virulence, and immune escape characteristics. Nonetheless, while not all future mutations will arise through the same mechanisms, the current study on the XDR lineage is still significant. Firstly, the study enhances our understanding of how SARS-CoV-2 evolves, offering insights into potential future changes. Additionally, while the humoral immunity of those infected would likely align with the JN* infections, cellular immunity, which is mediated by T cells, and target epitopes other than the ones in the Spike protein, might not. This difference in immune recognition could influence the evolutionary trajectory of SARS-CoV-2 evolution, especially in regions where the XDR lineage became highly prevalent. We briefly discuss this point in the new version of the manuscript.

2.3) *What can be an experimental proof than just statistical analysis for your study conclusion?*

Answer: In the study, the effective reproductive number (R_e) and the relative instantaneous reproductive number (R_{RI}) were used as close approximations of the viral fitness of the XDR, JN*, and XBB* lineages in the context of the Brazilian epidemic between November 2023 and February 2024. Based on the results, we concluded that XDR and JN* exhibited equivalent viral fitness, an expected outcome given the critical role of the Spike protein in determining SARS-CoV-2 transmissibility (Gobeil et al., 2021) and its similar presentation in both lineages. However, the study does not provide experimental validation of this result, a limitation that has been acknowledged in the current version of the manuscript. To provide evidence of these

results, however, one would need to test the immune escape and intrinsic transmissibility properties (Letko et al., 2020) of the XDR lineage and compared to those observed for the parental lineages. Specifically, to assess immune escape, it would be necessary to demonstrate, for instance, a reduction in the PRNT50 titer of XDR isolates against plasma from individuals with XBB* infections, mirroring the observations made with the JN.1 lineage (Yang et al., 2024).

References

- Gobeil, S. M.-C., Janowska, K., McDowell, S., Mansouri, K., Parks, R., Stalls, V., Kopp, M. F., Manne, K., Li, D., Wiehe, K., Saunders, K. O., Edwards, R. J., Korber, B., Haynes, B. F., Henderson, R., & Acharya, P. (2021). Effect of natural mutations of SARS-CoV-2 on spike structure, conformation, and antigenicity. *Science*, 373(6555). <https://doi.org/10.1126/science.abi6226>
- Letko, M., Marzi, A., & Munster, V. (2020). Functional assessment of cell entry and receptor usage for SARS-CoV-2 and other lineage B betacoronaviruses. *Nature Microbiology*, 5(4), 562–569. <https://doi.org/10.1038/s41564-020-0688-y>
- Tamura, T., Ito, J., Uriu, K., Zahradnik, J., Kida, I., Anraku, Y., Nasser, H., Shofa, M., Oda, Y., Lytras, S., Nao, N., Itakura, Y., Deguchi, S., Suzuki, R., Wang, L., Begum, M. M., Kita, S., Yajima, H., Sasaki, J., ... Sato, K. (2023). Virological characteristics of the SARS-CoV-2 XBB variant derived from recombination of two Omicron subvariants. *Nature Communications*, 14(1). <https://doi.org/10.1038/s41467-023-38435-3>
- Yang, S., Yu, Y., Xu, Y., Jian, F., Song, W., Yisimayi, A., Wang, P., Wang, J., Liu, J., Yu, L., Niu, X., Wang, J., Wang, Y., Shao, F., Jin, R., Wang, Y., & Cao, Y. (2024). Fast evolution of SARS-CoV-2 BA.2.86 to JN.1 under heavy immune pressure. *The Lancet Infectious Diseases*, 24(2), e70–e72. [https://doi.org/10.1016/s1473-3099\(23\)00744-2](https://doi.org/10.1016/s1473-3099(23)00744-2)

Re: Spectrum01193-24R1 (Rapid spread of the SARS-CoV-2 Omicron XDR lineage derived from recombination between XBB and BA.2.86 subvariants circulating in Brazil in late 2023)

Dear Dr. Ighor Arantes:

Your manuscript has been accepted, and I am forwarding it to the ASM production staff for publication. Your paper will first be checked to make sure all elements meet the technical requirements. ASM staff will contact you if anything needs to be revised before copyediting and production can begin. Otherwise, you will be notified when your proofs are ready to be viewed.

Sincerely,
Ujjwal Neogi
Editor
Microbiology Spectrum